# Thyroid Dysfunction and the Effect of Iodine-Deficient Parenteral Nutrition in Very Low Birth Weight Infants: A Nationwide Analysis of a Korean Neonatal Network Database

**DOI:** 10.3390/nu14153043

**Published:** 2022-07-25

**Authors:** JaeYoung Cho, JeongHoon Park, JungSook Yeom, JinSu Jun, JiSook Park, EunSil Park, Ji-Hyun Seo, JaeYoung Lim, Chan-Hoo Park, Hyang-Ok Woo

**Affiliations:** 1Department of Pediatrics, Gyeongsang National University Hospital, Jinju 52727, Korea; chojy17@gmail.com (J.C.); wagguruku@naver.com (J.P.); jsyeom@gnu.ac.kr (J.Y.); dldirlquf@hanmail.net (J.J.); espark@gnu.ac.kr (E.P.); seozee@gnu.ac.kr (J.-H.S.); pedneu@gnu.ac.kr (J.L.); howoo@gnu.ac.kr (H.-O.W.); 2Department of Pediatrics, Gyeongsang National University College of Medicine, Jinju 52828, Korea; aroma@gnu.ac.kr; 3Institute of Health Sciences, Gyeongsang National University, Jinju 52828, Korea; 4Department of Pediatrics, Gyeongsang National University Changwon Hospital, Changwon 51619, Korea

**Keywords:** parenteral nutrition, iodine, thyroid, L-thyroxine, very low birth weight infant

## Abstract

Background: To investigate the impact of nutritional iodine deficiency on thyroid dysfunction (TD) in very low birth weight (VLBW) infants, we analyzed the association between iodine-deficient parenteral nutrition (PN) and TD requiring L-thyroxine (TD-LT4). Methods: Data of VLBW infants were obtained from the Korean Neonatal Network registry. Factors including duration of PN were analyzed according to TD-LT4. Results: TD-LT4 occurred in 490 (8.7%) of 5635 infants, and more frequently occurred in infants requiring PN for ≥4 weeks (10.2%). PN ≥ 4 weeks was one of the risk factors for TD-LT4, with an odds ratio (OR) of 1.346, *p* = 0.002. However, multivariate analysis showed that TD-LT4 was more of a risk for infants that were small for gestational age (OR 2.987, *p* < 0.001) and for other neonatal morbidities such as seizures (OR 1.787, *p* = 0.002) and persistent pulmonary hypertension (OR 1.501, *p* = 0.039) than PN ≥ 4 weeks (OR 0.791, *p* = 0.080). Conclusions: Prolonged iodine-deficient PN might affect TD-LT4 in VLBW infants. However, the effect of nutritional iodine deficiency on TD-LT4 risk was less than that of SGA or severe neonatal morbidities in Korean VLBW infants.

## 1. Introduction

Thyroid dysfunction (TD) in preterm or very low birth weight (VLBW, weighed <1500 g at birth) infants is common (9~53.3%) [1,2,3] and is affected by a delayed surge of thyroid stimulating hormone (TSH), immature hypothalamic-pituitary-thyroid axis, loss of maternal contribution, comorbidities, and drugs related to non-thyroidal illnesses [4,5]. Patterns of TD are classified as congenital hypothyroidism (CH) or hypothyroidism with delayed thyroid stimulating hormone (TSH) elevation, subclinical hypothyroidism with persistent hyperthyrotropinemia, and transient hypothyroxinemia of prematurity (THOP), respectively [6,7]. L-thyroxine (LT4) is often administered (8–59%) to preterm infants with TD due to concern for developmental issues. However, a treatment consensus has not yet been achieved, especially for THOP [1,8]. Because of the diversity of TD and a lack of treatment consensus, the incidence of TD is variable, and a nationwide incidence and treatment strategy in preterm or VLBW infants is hitherto lacking.

Iodine is a critical nutrient for thyroid hormone synthesis, and thyroid hormones are essential in the development of the brain and physical growth in infancy. A dose of 1–10 μg/kg/day via parenteral nutrition (PN) or 11–55 μg/kg/day via enteral nutrition (EN) is currently recommended in preterm infants to avoid iodine deficiency [9,10]. Iodine deficiency or excess may affect thyroid function in preterm infants [11,12]. Treatment with drugs affecting thyroid function or exposure to excess iodine, such as iodinated contrast agents and topical iodine antiseptics, frequently occurred in preterm or VLBW infants, which induces downregulation of thyroid hormone synthesis [13]. The downregulation slowly recovered in preterm infants and led to a hypothyroid state [4]. Besides the iodine absorption from the agents, iodine deficiency can occur in preterm or VLBW infants because maternal supplementation ceased before accumulating as much iodine as in term infants. Infants receiving total PN are more vulnerable to iodine deficiency because most PN formulations and parenteral multi-trace mineral products manufactured in the world contain insufficient iodine for preterm infants [14,15,16]. There is no available parenteral product for iodine supplementation in Korea. Even when the preterm or VLBW infants achieve full EN, iodine supplementation may be still insufficient via preterm infant formula, the mother’s own milk, or human milk fortifier [17]. In Korea, iodine-rich seaweed soup has been frequently served to lactating mothers. Despite the high dietary iodine intake of lactating women, the median iodine concentration in mature breast milk (BM, 822 μg/L) from Korean lactating mothers may not fulfill the daily recommendations for partial enteral-fed preterm infants requiring PN for significant duration [18]. TD is common in preterm or VLBW infants, and previous studies suggested a relation to nutritional iodine insufficiency [11,19], but the nationwide report has been rare. Therefore, we investigated the nationwide incidence of TD requiring LT4 and the association with iodine-deficient PN in VLBW infants via the Korean Neonatal Network (KNN) registry.

## 2. Materials and Methods

### 2.1. Study Population

Data were extracted from the KNN registry, which is comprised of data collected prospectively based on a standardized protocol from 89 neonatal intensive care units (NICU) in South Korea since 2013. The KNN database provides clinical data for VLBW infants from birth to 3 years of age, and all data have been collected by trained experts working at NICUs registered by the KNN. The data collected by the KNN registry included demographic information, neonatal morbidities and related treatments during hospitalization, and growth-developmental status, chronic morbidities, and medications of LT4 and antiepileptic drugs (AEDs) at two follow-up (FU) visits, respectively. The two FU visits are scheduled at 18–24 months of corrected age (FU1) and 33–39 months of chronological age (FU2), respectively. A total of 11,294 VLBW infants born in 2013–2018 were registered in the KNN registry. In this study, 5635 infants were included, excluding 5659 infants due to discontinuation of follow-up including 1741 deaths, severe congenital anomalies, or missing LT4 prescribing information (Figure 1).

### 2.2. Definition

The data were collected based on the standardized protocol of the KNN. Gestational age (GA) was determined based on the last menstrual period. Prenatal steroid was collected as complete or incomplete administration of any corticosteroid to mother within a week before delivery. Chorioamnionitis was confirmed by pathologic examination of placenta. Prolonged rupture of membrane (PROM) was defined as rupture of amniotic membrane over 18 h before labor. Respiratory distress syndrome of newborn (RDS) was defined as ground glass opacity on chest X-ray. Persistent pulmonary hypertension of newborn (PPHN) and treatment within a week after birth were collected. PPHN was diagnosed based on echocardiographic findings fulfilled with the right to left shunt and elevated pulmonary vascular resistance in the absence of underlying cyanotic congenital heart disease. Patent ductus arteriosus (PDA) was defined with the left to right shunt via PDA on echocardiography. Ligation and therapeutic or prophylactic medication of ibuprofen, indomethacin, or other non-steroidal anti-inflammatory drugs (NSAIDs) were also collected by the KNN. Hypotension was defined as a mean arterial pressure (MAP) below that for the GA or a MAP < 30 mmHg within a week after birth, and medications for hypotension including inotropes such as dopamine, epinephrine, hydrocortisone, and others such as vasopressin were obtained. Bronchopulmonary dysplasia (BPD) was defined as the necessity of oxygen or respiratory support at 36 weeks of gestation or postnatal 28th day based on the severity-based definition for BPD of the National Institutes of Health consensus [20]. Intraventricular hemorrhage (IVH) and periventricular leukomalacia (PVL) were diagnosed via cranial ultrasonography (USG) or magnetic resonance imaging. Neonatal seizure was collected when there was a history of taking AEDs. Necrotizing enterocolitis (NEC) was defined as ≥stage 2b according to the modified Bell’s criteria. Sepsis was defined by a positive blood culture and antibiotic treatment for ≥5 days. Retinopathy of prematurity (ROP) was diagnosed and classified by an ophthalmologist at each hospital, and the worst stage of ROP during hospitalization and operative history were collected. When the infant could achieve EN of ≥100 mL/kg/day, Full EN (FEN) was defined and the date was collected. Severe congenital anomaly was determined as serious and life-threatening congenital anomaly based on the International Statistical Classification of Diseases and Related Health Problems 10th revision.

The KNN registry also collected chronic respiratory, ophthalmologic, or hearing problems, medication of AEDs or LT4, and growth/developmental status at FU1 and FU2, respectively.

Developmental scales were collected from the Bayley Scales of Infant and Toddler development (BSID) II, BSID III, the Korean Developmental Screening Test (K-DST), or the Korean Ages and Stages Questionnaire (K-ASQ), depending on the availability in each hospital. K-DST is a parent-reported screening test with 6 developmental domains of gross motor, fine motor, cognition, language, sociality, and self-help attributes. It was developed by experts in related fields as part of a National Health Screening System for Infants and Children in Korea. The cutoff points consist of scores corresponding to the −2 standard deviation (SD) and −1 SD of the domain scores and have been calculated based on the score of the K-DST population group. The cutoff point of <−2 SD corresponded to ‘recommendation for further evaluation’, and −2 to −1 SD corresponded to ‘need for follow-up’, respectively. The validity for predicting mental developmental delay in VLBW infants was as useful as that of BSID II [21]. K-ASQ is a questionnaire with 5 domains of gross motor, fine motor, communication, problem-solving, and personal-social attributes, which is a revised Korean version of the Ages and Stages Questionnaire in the US [22].

Since the results of thyroid function tests (TFTs) were unavailable from the KNN database, the past or present prescription of LT4 at FU1 or FU2 was determined as an indirect factor for TD requiring LT4 (TD-LT4) and inclusions were divided into two groups according to TD-LT4. To assess the risk of iodine-deficient PN on TD-LT4, demographic data including the GA at birth, birth weight (BW), sex, obstetric information, Apgar score (AS), neonatal morbidities, and medications such as inotrope, LT4, or AEDs were obtained. Because there is no manufactured product for parenteral iodine supplementation available in South Korea, PN and FEN duration were obtained to investigate the risk of iodine-deficient PN on TD-LT4 in VLBW infants. Prophylactic and therapeutic-treated PDA was considered one of clinical factors because the NSAIDs could affect duration of FEN or PN in VLBW infants [23]. Since inotropes, blood transfusion, and exposure to antiseptic procedures using povidone-iodine can affect thyroid function in VLBW infants [5,13,24], neonatal morbidities requiring inotropic agents, such as PPHN and hypotension, red blood cell (RBC) transfusion, and operations caused by hemodynamically significant PDA, spontaneous intestinal perforation (SIP), NEC, and ROP were also obtained and analyzed.

To investigate the outcome of TD-LT4, growth parameters including weight, height, head circumference (HC), and the developmental status at FU1 or FU2 were obtained. Assessment of growth was performed using the Fenton growth chart until a post-menstrual age (PMA) of 50 weeks, and the 2017 Korean Children and Adolescents Growth Standard at the time of FU1 and FU2, respectively. Small for gestational age (SGA) was defined as a BW below 10th percentile for GA and sex. Extra-uterine growth retardation (EUGR) was defined as weight, height, or HC of <10th percentile for PMA and sex at discharge, and growth retardation (GR) was defined as a weight, height, or HC < 5th percentile at FU1 or FU2, respectively [25]. Corrected and chronologic age were applied to assess the growth parameters at FU1 and FU2, respectively.

As one of the outcome parameters, cognitive scales were assessed from the developmental screening tests collected by the KNN. Cognitive delay was defined as Bayley II Mental Developmental Index (MDI) < 70, Bayley III cognitive composite scores < 85, or the cognitive domain of K-DST < −1 SD [26,27]. The result of K-ASQ was not included because the cognitive scale was not clearly distinguished from the domains. Quadriplegic cerebral palsy (CP), ipsilateral hemiplegia, or diplegia at FU1 or FU2 were also obtained to analyze the outcome. When an infant was registered at both FU1 and FU2, the latest information was selected.

### 2.3. Statistical Analysis

A continuous variable was reported as mean and SD and compared using independent *t* or Mann–Whitney U test according to Kolmogorov–Smirnov or Shapiro–Wilk test. A categorical variable was reported as number and percentage and compared using chi-square or Fisher’s exact test. To evaluate the risk factor of TD-LT4, univariate linear or non-linear logistic regression analysis was performed with each significantly different factor from the comparison according to TD-LT4. Multivariate logistic regression analyses were performed adjusted with GA and statistically significant factors from univariate analyses to evaluate the risk of PN for TD-LT4 and the associated outcome with TD-LT4. When multivariate analyses were performed, GA was used for adjustment with other significant factors rather than BW because of the correlation coefficient (*r* = 0.686, *p* < 0.001). SPSS version 25 (IBM, Armonk, NY, USA) was used for analyses. Two-sided *p* < 0.05 was significant.

## 3. Results

A total of 5635 VLBW infants were born at 28.6 weeks of gestation and weighed 1091.0 g. The infants required PN for 28.4 days and achieved FEN on day 24.1 after birth. The mean hospital duration was 75.5 days, and the infants discharged from the NICU at 39.4 weeks PMA. Follow-up information for the infants was collected at the mean of 36.5 months after birth. LT4 was administered to 490 (8.7%).

### 3.1. Comparisons of Clinical Factors According to Thyroid Dysfunction Requiring L-thyroxine

Demographic characteristics, obstetric histories, neonatal morbidities and related treatments during hospitalization, and growth-developmental outcomes were described and compared according to LT4 (Table 1). GA at birth (28.1 ± 2.45 vs. 28.7 ± 2.21 weeks, *p* <0.001) and BW (955.0 ± 275.2 vs. 1103.9 ± 256.2 g, *p* < 0.001) were lower in the LT4 (+) than the (−) group. The ratio of SGA (22.4% vs. 11.7%, *p* < 0.001) in the LT4 (+) group was higher than in the (−) group. Sex, obstetric history including maternal medical condition, use of prenatal steroids, delivery mode, pregnancy method, or birthplace was not different between the two groups. Poor AS at 5 min (0–3), multi-gestation, and PROM occurred more frequently in the LT4 (+) than in the (−) group. The infants in the LT4 (+) group required PN longer (34.2 ± 30.2 days) than those in the (−) group (27.9 ± 26.1 days, *p* < 0.001), and needed 6 days more to achieve FEN than those in the (−) group (29.7 ± 33.3 vs. 23.6 ± 21.4 days, *p* < 0.001), respectively. LT4 was administered more frequently in 220 (10.2%) of 2160 infants that required PN for ≥ 4 weeks than in 270 of 3475 infants requiring PN < 4 weeks (7.8%, *p* = 0.002). PPHN, medication for PDA, hypotension (with medications), seizure, sepsis, moderate to severe BPD, RBC transfusion, and operations occurred more frequently in the LT4 (+) than in the (−) group (*p* < 0.05). Of the long-term outcomes, GR of weight, height, or HC, cerebral palsy, cognitive delay, and AED were more prevalent in the LT4 (+) than in the (−) group (*p* < 0.05), respectively.

### 3.2. Risk Factors of Thyroid Dysfunction Requiring L-thyroxine

When univariate regression analyses were performed with significantly different factors from the comparisons between the two groups, the risk factors of TD-LT4 were as follows (Table 1); SGA (Odds ratio (OR) 2.193, *p* < 0.001), poor AS (OR 1.555, *p* = 0.026), multi-gestation (OR 1.293, *p* = 0.008), PPHN (OR 2.437, *p* < 0.001), moderate to severe BPD (OR 1.732, *p* < 0.001), drug for PDA (OR 1.564, *p* < 0.001), hypotension (OR 1.799, *p* < 0.001), seizure (OR 2.425, *p* < 0.001), PN ≥ 4 weeks (OR 1.346, *p* = 0.002), sepsis (OR 1.460, *p* < 0.001), operation (OR 1.903, *p* < 0.001), and RBC transfusion (OR 1.654, *p* < 0.001). GA was a statistically significant factor with TD-LT4, but the risk potential was weak (OR 1.124, *p* < 0.001).

When multivariate logistic regression analysis was performed with significant factors from univariate analyses, SGA weighed the most on the risk of TD-LT4 in VLBW infants (OR 2.987, *p* < 0.001). The others were listed as follows: seizure (OR 1.787, *p* = 0.002), RBC transfusion (OR 1.530, *p* = 0.019), PPHN (OR 1.501, *p* = 0.039), multi-gestation (OR 1.428, *p* = 0.003), drug for PDA (OR 1.348, *p* = 0.013), and hypotension (OR 1.315, *p* = 0.042), in order (Table 2). GA (per week decrease, OR 1.125, *p* < 0.001) and PN ≥ 4 weeks (OR 0.791, *p* = 0.080) were not significant risk factors of TD-LT4.

### 3.3. Long-Term Outcomes of Thyroid Dysfunction Requiring L-thyroxine

Infants in the LT4 (+) group stayed longer in the NICU than infants in the (−) group (90.8 ± 48.5 vs. 74.3 ± 35.6 days, *p* < 0.001). The ratios of EUGR at discharge and GR at FU were statistically significantly higher in the LT4 (+) than in the (−) group, and they were significantly associated with TD-LT4 on univariate analyses, respectively (Table 1). To investigate the associated long-term outcomes with TD-LT4, the outcome parameters were analyzed using multivariate regression analyses with significant neonatal morbidities from previous univariate analyses including GA (Table 3). TD-LT4 was not associated with CP (OR 1.084, *p* = 0.712) and cognitive delay (OR 1.165, *p* = 0.293), but was associated with GR (weight: OR 1.859, *p* <0.001, height: OR 1.449, *p* = 0.009, and HC: OR 1.595, *p* = 0.004).

## 4. Discussion

Iodine is an essential nutrient required for thyroid hormone synthesis. Thyroid hormones play a pivotal role in developing the brain and in physical growth in infants. VLBW infants often require PN for a significant period after birth until they achieve sufficient EN. However, iodine content in available PN formulations was negligible, and a parenteral iodine product for preterm infants was not available [15]. Additionally, iodine in preterm enteral formula is insufficient to meet requirements [17]. Even when the infant achieves EN of 100 mL/kg/day with preterm formula, iodine can be supplied 4.1–16.9 μg/kg/day (Appendix A). Therefore, prolonged PN may entail insufficient iodine supply, resulting in insufficient synthesis of thyroid hormones. In this study, prolonged PN for ≥ 4 weeks was one of the risk factors for TD-LT4 in VLBW infants on univariate analysis (OR 1.346, *p* = 0.002, Table 1). However, when multivariate analysis was performed, PN ≥ 4 weeks was found to be a protective factor for TD-LT4, albeit statistically insignificant (OR 0.791, *p* = 0.080, Table 2). The conflicting results might show other perinatal or neonatal morbidities statistically outweighed the risk of prolonged iodine-deficient PN for TD-LT4 in Korean VLBW infants. Among the factors, SGA was the greatest potential risk factor for TD-LT4 in Korean VLBW infants. Recent studies also reported that SGA infants at risk of TD and TSH were significantly higher in preterm SGA infants than in those of appropriate gestational age [28,29]. The cause of the TD in SGA infants is still being explored, but multifactorial mechanisms like uterine growth restriction, frequent non-thyroidal illness and related medications, caloric restriction, or less efficient thermogenesis are suggested [13,29]. Among non-thyroidal illnesses, seizures, PPHN, RBC transfusion, and related treatments for PDA and hypotension were the risk factors for TD-LT4 in this study. Multigestation is also one of the risk factors for TD-LT4 in this study. The previous preterm twin study reported that hyperthyrotropinemia was more prevalent in smaller than in larger twin infants [30]. Although risk of TD in twin infants was unknown, thyroid function might be affected by smaller BW secondary to multi-gestation, similar to SGA. However, since the KNN provided anonymized clinical data, the pairs of multigestation infants were veiled in this study.

VLBW infants with critical illness are often exposed to agents affecting thyroid function, such as povidone iodine, amidotrizoate, dopamine and its analogues, and glucocorticoid [5,13]. In this study, inotropes or glucocorticoid for hypotension (OR 1.283, 95% CI 0.987–1.667, *p* = 0.063) and operations as an indirect factor of povidone iodine (OR 1.165, 95% CI 0.880–1.541, *p* = 0.285) were not significantly associated with TD-LT4. Operations caused by PDA, SIP, NEC, or ROP were considered as an indirect factor of the exposure to povidone iodine in this study because detailed antiseptic procedures were not collected by the KNN registry. Although detailed antiseptic procedures were unknown, recent substitution of chlorhexidine for iodine disinfection in VLBW infants might reduce the risk of TD [31]. The effect of amidotrizoate on TD was not analyzed because information on meconium plug syndrome and related radiologic study or treatment was not available.

Partial EN with BM from mothers consuming a high-iodine diet while PN was required in VLBW infants might be another possible cause of reducing the deleterious effect of iodine-deficient PN on TD in Korea [32]. Unfortunately, however, the dietary types were not analyzed in this study because the type of EN formula was not collected by the KNN registry. Further research on the association between Korean BM and TD in VLBW infants is necessary.

Incidence of TD-LT4 was 8.7% in Korean VLBW infants (Figure 1). Previous studies reported the incidence of TD or TD-LT4 in preterm or VLBW infants was 12.2–30% at each center in Korea [1,2,33]. The different incidence was caused by differences in inclusion criteria and definitions from the previous studies, such as infants born at ≤32 weeks of gestation [1], or infants weighing < 1500 g [2]. The different definitions were set as congenital hypothyroidism, delayed TSH elevation, THOP, or subclinical hypothyroidism [33]. In western countries, the incidence of TD in this population has been reported to be 9.1–35.5%, depending on the definition and the timing of postnatal thyroid screening [6,34]. The rates of LT4 supplementation in VLBW infants with TD are also variable (8–59%) since the treatment consensus for THOP is unclear so far [4]. The incidence of TD-LT4 in this study might be underestimated because of unknown information about LT4 at FU1 or 2 from the infants supplemented with LT4 who lost contact before FU1 and infants untreated for TD.

GR at 2–3 years of age was associated with TD-LT4, but CP or cognitive delay was not (Table 3). When the associating factors were analyzed with the outcome parameters, SGA was the most potentially influential factor for GR of weight (OR 6.801, 95% CI 5.295–8.737, *p* < 0.001) and height (OR 4.939, 95% CI 3.877–6.291, *p* < 0.001). AEDs at 2–3 years of age were the greatest potential risk factor for CP (OR 26.923, 95% CI 13.99–51.82, *p* < 0.001) and cognitive delay (OR 10.625, 95% CI 4.027–28.03, *p* < 0.001), respectively (Appendix A). Although SGA was the greatest potential risk factor of GR, TD-LT4 was also a statistically significant risk (OR 1.918, *p* < 0.001) for GR despite treatment. The result may suggest that early detection and treatment of TD are necessary to reduce the risk of GR in VLBW infants.

There were several limitations. Lack of information regarding detailed results of thyroid function tests (TFTs) and agents affecting TD was the first. Since TFTs and patterns of TD were unavailable from the KNN database, we used information for LT4 prescriptions at FU1 or 2 as an indirect factor of TD requiring LT4. Because there were no detailed data on agents affecting thyroid function, hypotension treated with inotropes, glucocorticoids, or others, and operations were analyzed as indirect factors for drugs and disinfectant affecting thyroid function, respectively. Second, EN type and the effect according to the diet type could not be analyzed, especially in BM. Since iodine-rich seaweed soup has been frequently consumed by lactating mothers in Korea, iodine concentration in BM may be high and affect the thyroid function of the offspring [18]. Uncertain initiation time of LT4 and diversity in nutrition and treatment strategies depending on each NICU were other limitations.

Despite the limitations, several strengths could be suggested. We analyzed large-scale medical information from the nationwide database and could estimate the national incidence and risk factors of TD-LT4 in VLBW infants. Additionally, this study may be helpful in reconsidering the need for parenteral iodine supplementation.

In conclusion, based on the results, long-term iodine-deficient PN might affect TD, but, due to other risk factors, long-term iodine-deficient PN might not be necessary to relate to the requirement of L-thyroxine treatment for TD in VLBW infants in Korea. Apart from nutritional iodine supplementation, early detection and treatment of TD might be required, especially in SGA VLBW infants, since TD could affect long-term growth retardation despite treatment.

## Figures and Tables

**Figure 1 nutrients-14-03043-f001:**
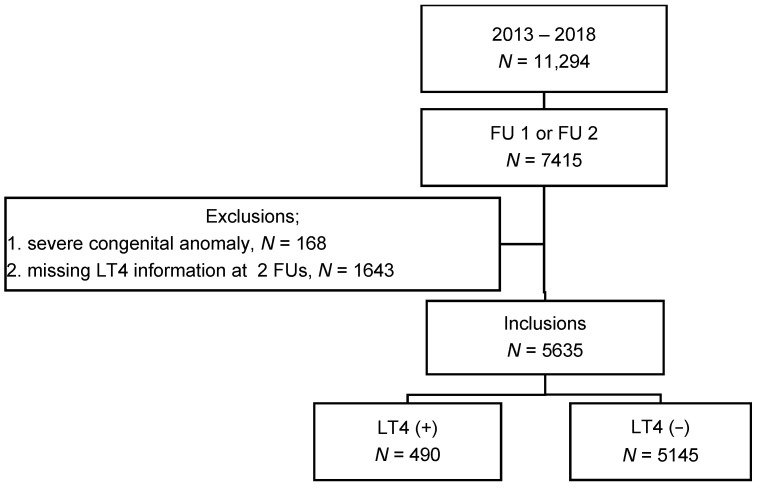
Number of inclusions and incidence of thyroid dysfunction requiring L-thyroxine supplementation. Abbreviations: FU1, follow-up visit at 18–24 months of corrected age; FU2, follow-up visit at 33–39 months of chronological age; LT4 (+), L-thyroxine supplementation; LT4 (−), no L-thyroxine supplementation.

**Table 1 nutrients-14-03043-t001:** Comparison and risk analysis of clinical characteristics in very low birth weight infants according to L-thyroxine administration for thyroid dysfunction.

Clinical Characteristics		Total (*N* = 5635)		LT4 (+) (*N* = 490)		LT4 (−) (*N* = 5145)		*p*	Odds Ratio	95% CI		**p*
		*N*, Mean	%, SD	*N*, Mean	%, SD	*N*, Mean	%, SD			Lower	Upper	
Demographic Factor												
Gestational age (weeks)		28.63	2.24	28.09	2.45	28.68	2.21	<0.001	1.124	1.079	1.171	<0.001
Birth weight (g)		1091.0	261.3	955.0	275.2	1103.9	256.2	<0.001	1.000	1.000	1.000	<0.001
Small for gestational age		710	12.6	110	22.4	600	11.7	<0.001	2.193	1.745	2.756	<0.001
Male		2837	50.3	234	47.8	2603	50.6	0.237				
Vaginal delivery		1207	21.4	115	23.5	1092	21.2	0.249				
5′ Apgar score (0–3)		269	4.8	32	6.6	237	4.6	0.016	1.555	1.055	2.290	0.026
In vitro fertilization		1371	24.3	132	26.9	1239	24.1	0.168				
Multigestation		2021	35.7	202	41.2	1809	35.2	0.009	1.293	1.071	1.562	0.008
maternal DM		549	9.7	36	7.3	513	10.0	0.066				
Maternal HTN		1189	21.1	102	20.8	1087	21.1	0.908				
Chorioamnionitis		1812	37.8	161	38.1	1651	37.8	0.916				
PROM		2090	37.3	160	33.1	1930	37.7	0.049	0.819	0.672	0.998	0.047
Prenatal steroid		4744	85.4	423	87.4	4321	85.2	0.201				
Outborn		173	3.1	22	4.5	151	2.9	0.073				
Neonatal Morbidities and Related Treatment												
Respiratory	RDS	4644	82.4	404	82.4	4240	82.4	1.00				
	PPHN	291	5.2	52	10.6	239	4.6	<0.001	2.437	1.778	3.341	<0.001
	BPD	1809	32.2	213	43.7	1596	31.1	<0.001	1.732	1.427	2.081	<0.001
Cardiovascular	Drugs for PDA	1960	48.5	208	58.6	1752	47.5	<0.001	1.564	1.254	1.950	<0.001
	Hypotension	1076	19.1	140	28.6	936	18.2	<0.001	1.799	1.460	2.215	<0.001
Neurologic	Seizure	310	5.5	55	11.2	255	5.0	<0.001	2.425	1.783	3.279	<0.001
	IVH ≥ 3	336	6.0	34	6.9	302	5.9	0.319				
	PVL	410	7.3	44	9.0	366	7.1	0.145				
Gastrointestinal/Nutritional	NEC ≥ st2	291	5.2	32	6.5	259	5.0	0.164	1.318	0.902	1.926	0.154
	PN ≥4 weeks	2160	38.3	220	44.9	1940	37.7	0.002	1.346	1.117	1.622	0.002
Others	Sepsis	1179	20.9	133	27.1	1046	20.3	0.004	1.460	1.183	1.801	<0.001
	Operation	1144	20.3	153	31.2	991	19.3	<0.001	1.903	1.553	2.332	<0.001
	RBC transfusion	3845	68.2	379	77.3	3466	67.4	<0.001	1.654	1.328	2.060	<0.001
Outcomes at Discharge and 2–3 Years of Age												
Discharge (EUGR)	Hospital stays (day)	75.5	37.1	90.8	48.5	74.3	35.6	<0.001	1.009	1.007	1.011	<0.001
	Weight (*n* = 5635)	3190	56.6	351	71.6	2839	55.2	<0.001	2.051	1.673	2.515	<0.001
	Length (*n* = 5205)	3393	65.2	363	78.6	3030	63.9	<0.001	2.073	1.647	2.609	<0.001
	HC (*n* = 5360)	2351	43.9	289	61.8	2062	42.2	<0.001	2.216	1.824	2.693	<0.001
FU1 or FU2 (GR < 5 p)	Weight (*n* = 5251)	840	16.0	146	30.5	694	14.5	<0.001	2.575	2.086	3.178	<0.001
	Height (*n* = 4904)	894	18.2	138	29.7	756	17.0	<0.001	2.055	1.660	2.565	<0.001
	HC (*n* = 4164)	643	15.5	107	27.0	536	14.3	<0.001	2.219	1.746	2.821	<0.001
	AED (*n*= 5093)	69	1.2	11	2.3	58	1.1	0.047	2.032	1.059	3.898	0.033
FU1 or FU2 (Developmental delay)	Cerebral palsy (*n* = 4987)	345	6.3	46	9.6	299	6.0	0.004	1.658	1.197	2.296	0.002
	Cognitive delay (*n* = 3946)	1165	29.5	140	36.9	1025	28.7	0.001	1.453	1.165	1.812	0.001

*p* value was obtained by Fisher’s exact test or Mann–Whitney U test. **p* value was obtained by univariate linear or non-linear logistic regression analysis. Abbreviations and meanings: LT4, L-thyroxine; Multigestation, two or more fetuses in a pregnancy; Maternal DM, gestational diabetes or overt diabetes mellitus of mother; Maternal HTN, pregnancy-induced hypertension or essential hypertension; PROM, premature rupture of membrane; Prenatal steroid, completed or incomplete corticosteroid injection to mother before giving birth; RDS, respiratory distress syndrome; PPHN, persistent pulmonary hypertension; BPD, bronchopulmonary dysplasia; Drugs for PDA, ibuprofen, indomethacin and other nonsteroid anti-inflammatory drugs for closing patent ductus arteriosus therapeutically or prophylactically; Hypotension, hypotension (a mean arterial pressure (MAP) below that for the gestational age or a MAP < 30 mmHg within a week after birth) treated with medications including inotropes such as dopamine, dobutamine, or epinephrine, hydrocortisone, and others; Seizure, neonatal seizure requiring antiepileptic drugs; IVH ≥ 3, intraventricular hemorrhage ≥ grade 3; PVL, periventricular leukomalacia; PN ≥ 4 weeks, parenteral nutrition duration of ≥ 4 weeks; Sepsis, antibiotics for ≥ 5 days and positive result of blood culture; Operation, ligation of PDA, laparotomy of spontaneous intestinal perforation or NEC, or surgery for ROP; RBC transfusion, red blood cell transfusion; EUGR, extrauterine growth retardation; HC, head circumference; FU, follow up; GR < 5 p, growth retardation < 5th percentile of weight, height or HC at FU1 or 2; AED, antiepileptic drug; Cerebral palsy, any types of quadriplegia, diplegia or hemiplegia; Cognitive delay, Bayley II Mental Developmental Index (MDI) < 70, Bayley III cognitive composite scores < 85, or the cognitive domain of K-DST < −1 SD.

**Table 2 nutrients-14-03043-t002:** Risk factors of thyroid dysfunction requiring L-thyroxine in very low birth weight infants.

Risk Factors	Odds Ratio	95% CI		*p*
		Lower	Upper	
SGA	2.987	2.215	4.028	<0.001
Seizure	1.787	1.229	2.598	0.002
RBC transfusion	1.530	1.072	2.184	0.019
PPHN	1.501	1.020	2.210	0.039
Multigestation	1.428	1.133	1.798	0.003
Drugs for PDA closure	1.348	1.066	1.705	0.013
hypotension	1.315	1.010	1.713	0.042
PN ≥ 4 weeks	0.791	0.608	1.028	0.080
GA, per-week decrease	1.125	1.057	1.197	<0.001

*p* values were obtained by multivariate stepwise backward logistic regression analysis. Abbreviations: SGA, small for gestational age; RBC transfusion, red blood cell transfusion; PPHN, persistent pulmonary hypertension; Multigestation, two or more fetuses in a pregnancy; Drugs for PDA, ibuprofen, indomethacin and other nonsteroid anti-inflammatory drugs for closing patent ductus arteriosus therapeutically or prophylactically; PN ≥ 4 weeks, parenteral nutrition duration of ≥4 weeks; GA, gestational age.

**Table 3 nutrients-14-03043-t003:** Outcomes and influences of thyroid dysfunction requiring L-thyroxine in very low birth weight infants at 2 to 3 years of age.

Outcomes		Odds Ratio	95% CI		*p*
			Lower	Upper	
Growth retardation	Weight	1.918	1.455	2.528	<0.001
	Height	1.464	1.108	1.934	0.007
	Head circumference	1.636	1.191	2.246	0.002
Development	Cerebral palsy	1.095	0.714	1.679	0.676
	Cognitive delay	1.162	0.875	1.543	0.299

*p* values were obtained by multivariate logistic regression analyses adjusted with gestational age and differentiating clinical factors from univariate analyses. Growth retardation was defined as a weight, height, or head circumference < 5th percentile at FU1 or FU2. Cognitive delay was defined as Bayley II Mental Developmental Index (MDI) < 70, Bayley III cognitive composite scores < 85, or the cognitive domain of K-DST < −1 SD. Cerebral palsy was defined as quadriplegic, ipsilateral, and diplegic.

## Data Availability

Data is contained within the article and Appendix A.

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
