# Peer review of "Thyroid Dysfunction and the Effect of Iodine-Deficient Parenteral Nutrition in Very Low Birth Weight Infants: A Nationwide Analysis of a Korean Neonatal Network Database"

_nutrients, 2022, doi:10.3390/nu14153043_

Round 1

Reviewer 1 Report

This study uses a national database to explore thyroid dysfunction and the effect of iodine-deficient parenteral nutrition among a mumble of perinatal factors in Very Low Birth Weight Infants. In some way, the results need to be improved, the discussion needs to be rewritten, and the conclusions are not supported by the results. More efforts are encouraged for a good publication.

Author Response

  1. Following the reviewer’s comments, we tried to revise the result clearly and the conclusion along with the results in the abstract and manuscript.

-Abstract

 Conclusions: Prolonged iodine-deficient PN might affect TD-LT4 in VLBW infants. However, the risk of nutritional iodine deficiency on the TD-LT4 was less than that of SGA or severe neonatal morbidities in Korean VLBW infants.

- Conclusion

Despite the limitations, several strengths could be suggested. We analyzed large-scaled medical information from the nationwide database and could estimate the national incidence and risk factors of TD-LT4 in VLBW infants. Additionally, this study may be helpful in reconsidering the need for parenteral iodine supplementation.

In conclusion, based on the results, long-term iodine-deficient PN might affect TD, but due to other risk factors, intravenous iodine supplementation in PN might not be necessary to prevent TD in VLBW infants in Korea.

2. Since this study was performed using indirect information on risk factors for thyroid dysfunction requiring LT4 in VLBW infants, including duration of PN, there was a limit to deriving direct results regarding on relation between iodine-deficient PN and TD. This was the limitation of this study, and the limitation was explained in the discussion. However, although indirect, this study was meaningful because nationwide data was analyzed.

Reviewer 2 Report

The paper entitled "Thyroid dysfunction and the Effect of Iodine-deficient Parenteral Nutrition in Very Low Birth Weight infants: A Nationwide Analysis of a Korean Neonatal Network Database " is well written and clearly presented. However,  in the Introduction section, the authors need to describe what they consider a"Very Low Birth Weight infants" category. 

Minor: Authors need to decipher FU1 and FU2 abbreviations in the legend of figure1

Author Response

  1. As the reviewer's comment, we described the category of VLBW infants in the introduction. => Thyroid dysfunction (TD) in preterm or very low birth weight (VLBW, weighed <1,500 g at birth) infants is common (9~53.3%) [1–3]
  2. FU1 and FU2 abbreviations were explained in the legend of figure 1. => 

    Figure 2. Number of inclusions and Incidence of thyroid dysfunction requiring L-thyroxine supplementation. Abbreviations: FU1, Follow-up visit at 18-24 months of corrected age; FU2, Follow-up visit at 33-39 months of chronological age; LT4 (+), L-thyroxine supplementation; LT4 (-), no L-thyroxine supplementation.

Round 2

Reviewer 1 Report

The updated manuscript has made improvements.

The data from Table 2 shows that, among several factors, PN ≥ 4 weeks is the only possible protective factor of thyroid dysfunction requiring L-thyroxine in Very Low Birth Weight Infants(OR: 0.791, 95% CI 0.608 - 1.028, P = 0.080), while the data from Table 1 shows that, PN ≥ 4 weeks was one of risk factors (OR: 1.346, P = 0.002).

These results may be one of the vital information of this study. Thus, the authors need more stress on this information both in the results and discussion section; even the authors suspected two factors of exposure to agents affecting thyroid function, and partial parenteral nutrition with breast milk from mothers consuming high iodine diet.

While more studies in this area are warranted, are there already studies in this area available? Also, the authors are encouraged to analyze with different models to explore their connections.

Meanwhile, too many abbreviations in the context (such as VD, vaginal delivery) make the manuscript challenging to understand. Also, there are some bugs in language consideration, such as

The mean hospital duration were (was) 75.5 days ,,,,

at FU1 or FU2 vs. at FU1 or 2; 

Regarding the conclusions, the authors did not directly explore the effects of intravenous iodine supplementation in PN in this study. Thus, the conclusions should be driven with caution.

The following statement may be more prudent for the authors’ reference:

In conclusion, based on the results, long-term iodine-deficient PN might affect TD, but, due to other risk factors, long-term iodine-deficient PN might not be necessary to relate to the requirement of L-thyroxine treatment in VLBW infants in Korea….

Author Response

Comments and Suggestions for Authors

The updated manuscript has made improvements.

  1. The data from Table 2 shows that, among several factors, PN ≥4 weeks is the only possible protective factor of thyroid dysfunction requiring L-thyroxine in Very Low Birth Weight Infants (OR: 0.791, 95% CI 0.608 - 1.028, P= 0.080), while the data from Table 1 shows that, PN ≥ 4 weeks was one of risk factors (OR: 1.346, P = 0.002). These results may be one of the vital information of this study. Thus, the authors need more stress on this information both in the results and discussion section; even the authors suspected two factors of exposure to agents affecting thyroid function, and partial parenteral nutrition with breast milk from mothers consuming high iodine diet.
  • Following the reviewer’s comment, we added some opinions regarding the conflicting results as follows, in discussion: In this study, prolonged PN for ≥ 4 weeks was one of the risk factors for TD-LT4 in VLBW infants on univariate analysis (OR 1.346, P = 0.002, Table 1). However, when multivariate analysis was performed, PN ≥ 4 weeks was found to be a protective factor for TD-LT4, albeit statistically insignificant (OR 0.791, P = 0.080, Table 2). The conflicting results might show other perinatal or neonatal morbidities statistically outweighed the risk of prolonged iodine-deficient PN for TD-LT4 in Korean VLBW infants.

  1. While more studies in this area are warranted, are there already studies in this area available?
  • There are several studies in nutritional iodine deficiency in preterm infants, which I’ve already cited in the introduction and discussion (ref 14~19). However, there is a lack of reports regarding a nationwide association between nutritional iodine deficiency and TD in VLBW infants.

  1. Also, the authors are encouraged to analyze with different models to explore their connections.
  • As the reviewer’s comment, we discussed a different model to analyze the complex confounding factors with the medical research statistics of GNUH and they advised the multivariate logistic regression analysis was appropriate to analyze the risk factors of TD-LT4.

  1. Meanwhile, too many abbreviations in the context (such as VD, vaginal delivery) make the manuscript challenging to understand. Also, there are some bugs in language consideration, such as The mean hospital duration were (was) 75.5 days ,,,, at FU1 or FU2 vs. at FU1 or 2;
  • As the reviewer’s comment, several abbreviations in Table1 were revised.
  • We fixed the language bugs in the results and tables.

  1. Regarding the conclusions, the authors did not directly explore the effects of intravenous iodine supplementation in PN in this study. Thus, the conclusions should be driven with caution. The following statement may be more prudent for the authors’ reference: In conclusion, based on the results, long-term iodine-deficient PN might affect TD, but, due to other risk factors, long-termiodine-deficient PN might not be necessary to relate to therequirement of L-thyroxine treatment in VLBW infants in Korea
  • Following the reviewer’s recommendation, we revised the sentence as follows; In conclusion, based on the results, long-term iodine-deficient PN might affect TD, but, due to other risk factors, long-term iodine-deficient PN might not be necessary to relate to the requirement of L-thyroxine treatment for TD in VLBW infants in Korea.
